# Transdermal Delivery of Lidocaine-Loaded Elastic Nano-Liposomes with Microneedle Array Pretreatment

**DOI:** 10.3390/biomedicines9060592

**Published:** 2021-05-23

**Authors:** Yang Liu, Maosen Cheng, Junqi Zhao, Xiaoying Zhang, Zhen Huang, Yuhui Zang, Ying Ding, Junfeng Zhang, Zhi Ding

**Affiliations:** 1State Key Laboratory of Pharmaceutical Biotechnology, School of Life Sciences, Nanjing University, Nanjing 210023, China; MG1830099@smail.nju.edu.cn (Y.L.); MG1930097@smail.nju.edu.cn (M.C.); MG1930111@smail.nju.edu.cn (J.Z.); MF1430031@smail.nju.edu.cn (X.Z.); zhenhuang@nju.edu.cn (Z.H.); zangyh@nju.edu.cn (Y.Z.); jfzhang@nju.edu.cn (J.Z.); 2Department of Anesthesiology, The Second Affiliated Hospital of Nanjing Medical University, Nanjing 210011, China; 3Changzhou High-Tech Research Institute of Nanjing University, Changzhou 213164, China

**Keywords:** lidocaine, solid microneedles, elastic vesicles, topical drug administration, anesthesia

## Abstract

This study aimed to improve the transdermal delivery of lidocaine hydrochloride (LidH) using elastic nano-liposomes (ENLs) and microneedle (MN) array pretreatment. LidH-containing ENLs were prepared using soybean phosphatidylcholine and cholesterol, with Span 80 or Tween 80, using a reverse-phase evaporation method. The ENL particle size, stability, and encapsulation efficiency (EE) were characterized and optimized based on the component ratio, pH, and type of surfactant used. In vitro transdermal diffusion study was performed on MN-pretreated mouse skin using Franz diffusion cells. The anesthetic effects of LidH in various formulations after dermal application were evaluated in vivo in rats by measuring the tail withdrawal latency after photothermic stimulation. Stable LidH-loaded Tween 80 or Span 80 ENLs were obtained with particle sizes of 115.8 and 146.6 nm and EEs of 27% and 20%, respectively. The formulations did not exert any cytotoxicity in HaCaT cells. Tween 80 and Span 80 ENL formulations showed enhanced LidH delivery on pretreated mice skin in vitro and prolonged the anesthetic effect in vivo compared to that by LidH application alone. LidH-loaded ENLs applied to MN-pretreated skin can shorten the onset time and prolong the anesthetic effect safely, which merits their further optimization and practical application.

## 1. Introduction

Lidocaine hydrochloride (LidH) is the only intravenous anesthetic approved by the Food and Drug Administration [1]. As a local anesthetic, lidocaine blocks nerves via the inhibition of the voltage-gated sodium, potassium, and calcium channels and other receptors [2]. Currently, LidH is mainly administered by intramuscular injection or transdermal delivery using mucilages; however, this method of delivery has some shortcomings in clinical application. For example, anesthesia for surgical operations that involve superficial skin, such as esthetic surgeries, tattooing, birthmark removal, scar dispelling, and skin transplantation, often require multiple injections, thereby causing pain and discomfort and generating a large amount of sharp contaminants [3]. Methods for transdermal delivery commonly take a long onset time (e.g., 60 min for the eutectic mixture of the local anesthesia EMLA^TM^, a commercial topical anesthesia cream that contains 2.5% lidocaine and 2.5% prilocaine), which prolongs the preparation time for surgery [4,5]. Methods to enhance LidH transdermal delivery have been developed using iontophoresis [6], sonication [7], and laser pretreatment [1] of the skin, but these methods need sophisticated equipment and can damage skin permanently. Thus, improving the transdermal delivery of topical anesthetics featuring economy and more safety, while shortening the onset time and prolonging anesthesia duration, will benefit both patients and physicians. Drug delivery systems like microparticles, ethosomes, solid lipid nanoparticles (SLNs), and elastic nano-liposomes (ENLs) can increase skin transportation [8]. Among them, microparticles present higher particle size and poor skin permeation than SLNs and ENLs; and ethosomes lack long-term structural and chemical stability during storage, following by possibility of skin irritation due to high ethanol content [8].

ENLs, also known as transfersomes or deformable liposomes, have been optimized for transdermal drug delivery [9]. ENLs possess higher elasticity and smaller vesicle size than SLNs, the traditional liposomes without surfactants, because of the surfactants addition [8]. El Maghraby et al., prepared Span 80- and Tween 80-containing ENLs encapsulating estradiol using a thin-film hydration method [10]. In vitro permeation studies have shown that the transepidermal fluxes of ENLs containing Span 80 and Tween 80 are approximately six and nine times higher than those of SLNs, respectively [11]. The use of ENLs alone for topical anesthesia application increases drug deposition in the *stratum corneum* and dermis [12]. Although liposomes enhance the transdermal delivery of drugs to some extent, the barrier function of the *stratum corneum* reduces the transdermal permeation of drugs, which prevents its therapeutic window from being reached.

Microneedle (MN) arrays, an arrangement of needles shorter than 1 mm, mechanically perforate the *stratum corneum* and form conduits for transdermal drug delivery [13,14]. The topical application of drugs using MN arrays may optimize the drug delivery across the skin and enhance its therapeutic effects [15]. The MNs do not touch the nerve endings or the capillaries of the dermis, as they can only reach the epidermis [16]. Therefore, MN application causes little pain and no bleeding, thereby reducing the risk of discomfort and infection accompanied by intramuscular injection [17]. MNs are usually classified as solid MNs, drug-coated MNs with the drug loaded onto the MN surface, hollow MNs with internal conduits, and dissolvable MNs with the drug incorporated into a soluble matrix [18,19,20,21]. As the first-generation, solid MNs are convenient for animal experiments and capable of enhancing transdermal LidH delivery.

To the best of our knowledge, the combined strategy of using solid MN array pretreatment followed by the application of LidH-loaded ENLs has not been reported. Thus, the purpose of this study was to shorten the onset time and prolong the duration of local anesthesia using LidH-loaded ENLs after pretreating the skin with a solid MN array. This study describes the preparation, characterization, optimization, and in vitro toxicological properties of LidH ENLs. Additionally, the transdermal diffusion characteristics were investigated in vitro, and the pharmacodynamics of LidH ENLs after MN pretreatment were assessed in rats (Figure 1). The results demonstrate that the combination of MN array pretreatment and ENLs formulations can enable local anesthetization more rapidly and economically, in a safe manner.

## 2. Materials and Methods

### 2.1. Materials

Dulbecco’s modified Eagle’s medium (DMEM) was purchased from Gibco^®^ Life Technology (Grand Island, NY, USA). Polyoxyethylene sorbitan monooleate (Tween 80) and sorbitan monooleate (Span 80) were obtained from J&K^®^ (Shanghai, China). Soybean phosphatidylcholine (SPC) and cholesterol were purchased from Shanghai Advanced Vehicle Technology Co., Ltd. (Shanghai, China) and the Cell Counting Kit-8 (CCK-8) was obtained from Dojindo Co., Ltd. (Shanghai, China). Red blood cell lysis buffer was purchased from Beyotime Biotech (Shanghai, China). LidH was purchased from Aladdin Co., Ltd. (Shanghai, China). Protamine sulfate and rhodamine B were obtained from Sangon Biotech Ltd. (Shanghai, China). Pentobarbital sodium was obtained from Merck (Darmstadt, Germany). All chemicals used were of analytical grade, and water was double distilled.

### 2.2. Animals

Specific pathogen-free BALB/c mice and Sprague Dawley rats (female, 6–8 weeks old at the beginning of the experiments) were obtained from the Laboratory Animal Center of Nanjing Medical University (Nanjing, China) and maintained under standardized pathogen-free conditions in the animal facility of the State Key Laboratory of Pharmaceutical Biotechnology, Nanjing University. All mice and rats were housed at a constant temperature (22 ± 1 °C) and humidity (55 ± 10%) under a 12 h light/dark cycle with free access to food (pelleted feed) and water. All animals were housed in standard plastic cages with corncob granule bedding (four mice or three rats per cage) and acclimatized for 1 week before further experiments. The animal use protocols have been reviewed and approved by the Animal Ethical and Welfare Committee of Nanjing University (IACUC-2007017).

### 2.3. LidH SLNs and ENLs Preparation

The LidH SLNs and ENLs were prepared using the reverse-phase evaporation method as previously described [22]. Briefly, SPC and cholesterol in the mass ratio 5:1 were dissolved in trichloromethane and mixed evenly. Two surfactants, Span 80 or Tween 80, were chosen and weighed according to the SPC: surfactant mass ratio of 10:1 in the ENL formulations, while no surfactant was added in the SLN formulation [23]. LidH solution (2%, *w*/*w*) was added to the mixture (1:3, *v*/*v*). The mixture was sonicated in an ice-bath for 3 min, until the dispersion became milky and monophasic. If the dispersion was not layered after at least 30 min, the solvent was removed from the reverse micelle dispersion by rotary evaporation under reduced pressure (200 rpm, 45 °C, 355 mmHg). After a thin and viscous film formed on the inner wall of the round-bottom flask and collapsed into a suspension, 1× phosphate-buffered saline (PBS; 10 mM phosphate buffer, 137 mM NaCl, pH 7.4) was added to hydrate the suspension under reduced pressure. Finally, the emulsion was sonicated for 3 min in an ice-bath, under 3 s sonication with 5 s intervals.

### 2.4. Determination of Entrapment Efficiency (EE)

The amount of LidH entrapped in liposomes was illustrated using the protamine aggregation method. Protamine solution (0.1 mL, 10 mg/mL) was added to 1.5 mL Eppendorf tubes containing 0.1 mL of liposome suspension and mixed for 3 min. After the addition of physiological saline (0.9% *w*/*w*, 1 mL), the supernatant was obtained from the mixture by centrifugation for 20 min at 48,400× *g* at 20 ± 5 °C. The amount of free LidH (W1) was measured using a UV spectrophotometer (Persee, TU-1800, Beijing, China) at 228 nm. The LidH content of the stock solution (W2) was determined by ethanol demulsification of the liposome stock solution (0.1 mL). EE was calculated using the following equation:% EE = (W_2_ − W_1_)/W_2_ × 100% (1)

### 2.5. Formulation Characterization

To obtain the mean particle size and polymer dispersity index (PDI), the LidH liposome dispersion was diluted to 0.1 mg/mL with distilled water, mixed evenly, and measured three times with a particle size analyzer (Brookhaven, 90 PLUS, Austin, TX, USA).

### 2.6. Cytotoxicity Test

The human immortalized keratinocyte cell line HaCaT was purchased from the Cell Bank of the Chinese Academy of Sciences (Shanghai, China), and cultured in DMEM (90%) supplemented with fetal bovine serum (FBS, 10%) and penicillin (100 U/mL)-streptomycin (100 μg/mL) in an incubator (Thermo Scientific, Shanghai, China) at 37 °C and 5% CO_2_. The cell suspensions (~5 × 10^5^ cells/mL) were seeded in 96-well plates (100 μL/well) and cultured for 24 h. Then, different concentrations (12.5, 25, 50, and 100 μg/mL) of Span 80 ENL (10 μL) or Tween 80 ENL (10 μL) were added to the wells and cultured for 6 h. Cell viability was measured using the CCK-8 assay according to the manufacturer’s instructions.

### 2.7. Skin Insertion of MN

To evaluate the skin penetration by the MN array, mice (*n* = 3) were anesthetized with 2% pentobarbital sodium by intraperitoneal injections (40 mg/kg) and the abdominal skin of mice was pretreated with the MNs. Each MN was assembled as a 500 μm-long, 6 × 6 array on a backplate. Then, 1% trypan blue solution (*w*/*w*) was smeared onto the skin and the treated mice were kept on a temperature-controlled mat for 30 min at 37 °C. The mice were euthanized by cervical vertebra dislocation and the skin was photographed under an inverted microscope (Nikon, Ti-U, Tokyo, Japan).

Similarly, rhodamine B (1 mg/mL) was applied to the abdominal skin of mice (*n* = 3) with or without MN array pretreatment and the mice were also kept at 37 °C for 30 min on the temperature-controlled mat. Then, the mice were euthanized by cervical vertebra dislocation and the skin was cut into small pieces (1 × 1 cm^2^), frozen, and sectioned. The sections were imaged and analyzed by laser scanning confocal microscopy (Ti-C2, Nikon, Tokyo, Japan).

### 2.8. Transdermal Diffusion of LidH from ENLs In Vitro

Less than 1 h after the mice (*n* = 3) were euthanized by cervical vertebra dislocation, the abdominal skin (2 × 2 cm^2^) was removed, examined for wounds, and pierced with the MN array (Figure 2a). A piece of skin with or without MN array pretreatment was placed on the top of the receptor compartment containing 1× PBS solution (15.5 mL) in a Franz diffusion cell. Next, 100 μL of the LidH solution, SLN or ENL formulation was spread evenly on the area of treated skin. The occlusive application was conducted with occluded donor compartments sealed with parafilm, while the non-occlusive application was conducted with open donor compartments. Once preparation was completed, the cells were placed on a transdermal diffusion system (Xinzhou, TP-6, Tianjin, China) at 37 °C with 350 rpm stirring. Samples (400 μL) were removed from the receptor compartment at 15 min (0–1 h), 30 min (1–3 h), and 1 h (3–6 h) intervals. After each removal, the same amount of prewarmed PBS was added. The concentration of LidH was measured using a UV spectrophotometer (Persee, TU-1800, Beijing, China).

### 2.9. In Vivo Thermal Hyperalgesia Study

To study the effect of local anesthesia, the thermal pain threshold of rat tails was determined as previously described [24]. The D’Armour and Smith test was employed using a tail flick test apparatus of a thermal radiation stimulator (Sansbio, SA-YLS-12A, Nanjing, China) for pain induction. The tail-flick reaction time of rats was measured as an index of pain sensitivity. The rats were assigned randomly into seven groups (*n* = 5): (1) blank, (2) LidH, (3) MN/LidH, (4) Span 80 ENL, (5) MN/Span 80 ENL, (6) Tween 80 ENL, and (7) MN/Tween 80 ENL. The rats were fixed in a glass tube, with only their tails exposed. After the rats were allowed to settle for 30 min, the middle part of the tail was exposed to thermal radiation through the stimulator; the protective time was set to 20 s to avoid excessive injury. The rat tail withdrawal latency (TWL) was recorded each time after the irradiation at 5 min intervals for 30 min. Then, after alcohol application and volatilization, the following formulations with LidH of 3.75 mg/cm^2^ were administered to a 1 cm^2^ area of the photothermic stimulation site per group (except the blank group), and incubated for 15 min in an occlusive dressing using sterile bandages. In detail, 187.5 μL LidH solution was applied to groups 2 and 3, and 526.5 μL ENL dispersion was applied to groups 4–7.

### 2.10. Statistical Analysis

All results are presented as the mean ± SD. Statistical analysis was carried out using a two-sided Student’s *t*-test or ANOVA with Bonferroni’s post-hoc test in the software GraphPad 7.0 (Prism, San Diego, CA, USA). A *p*-value of <0.05 was considered significant.

## 3. Results

### 3.1. Preparation and Optimization of LidH-Loaded ENLs

The preparation method used was previously verified as efficient to introduce diverse agents into organic systems in vitro and in vivo [22]. The influence of surfactants was evaluated, as no surfactant, Span 80, or Tween 80 was added in the system. The sizes of the liposomes are shown in Figure 3a. The average particle sizes of Span 80 and Tween 80 ENLs were 146.6 and 115.8 nm, respectively. The PDI of SLNs, Span 80, and Tween 80 ENLs were 0.284, 0.277, and 0.236, respectively. Tween 80-containing ENLs displayed lower PDI and higher homogeneity than SLNs (Figure 3b). The EEs of SLN, Tween 80 ENL, and Span 80 ENL were 31%, 27%, and 20%, respectively. The results showed that all groups achieved high EE levels. Significant differences in EE were only observed between the Tween 80 ENL and SLN groups (*p* < 0.05; Figure 3c). The EE of LidH in the ENLs was slightly lower than that in the SLNs.

HaCaT cells were used to evaluate the cytotoxicity of ENLs (Figure 3d). The average cell survival rates were approximately 98.75% and 98.05% 3 h after Span 80 ENL and Tween 80 ENL addition. No group showed significant cytotoxicity up to an ENL concentration of 100 μg/mL. 

### 3.2. MN Pretreatment of the Skin

Figure 4a shows the skin area of the mouse for MN pretreatment. The conduits formed were observable after incubation with 1% trypan blue (Figure 4b). The untreated and MN-pretreated skin were frozen sectioned for histological observation. Unlike in the untreated skin (Figure 4c), MNs pierced the *stratum corneum* and formed conduits in the treated skin (Figure 4d); thus, LidH could permeate through the skin faster after MN pretreatment. The amounts of LidH that permeated the skin in the SLN, Span 80 ENL, and Tween 80 ENL groups with MN pretreatment were 1.62, 1.86, and 2.40 times higher than those in the untreated groups, respectively (Figure 4e–g). The results showed that transdermal LidH flux was significantly increased by MN pretreatment (*p* < 0.05), especially in the Tween 80 ENL group (*p* < 0.01). 

### 3.3. Transdermal Delivery of LidH from ENLs on MN-Pretreated Skin In Vitro

To increase the transdermal flux of LidH and shorten the lag time, formulations were applied to the skin using an occlusive dressing. After 6 h of incubation, the permeated amount of LidH in the occlusive application group increased by approximately 80.2% from 539.2 ± 106 to 971.7 ± 67 μg/cm^2^ compared with that in the LidH solution group and by approximately 21.3% from 539.2 ± 106 to 654.1 ± 89 μg/cm^2^ compared with that in the non-occlusive application group (Figure 2b). 

In this assay, the occlusive topical application of LidH solution, LidH-loaded SLNs, Span 80 ENL, and Tween 80 ENL with MN pretreatment were investigated, using LidH solution on intact skin as the control. The cumulative amount of permeated LidH in both ENL groups was significantly higher than that of the control group (*p* < 0.001), and that in the SLN group was significantly lower than that in the Tween 80 ENL (*p* < 0.01) and Span 80 ENL (*p* < 0.05) groups. After MN treatment, the cumulative amount of permeated LidH in the SLN, Span 80 ENL, and Tween 80 ENL groups was 1.21, 1.67, and 2.1 times higher than that of the MN/LidH group, respectively (Figure 2c). In summary, the transdermal flux of LidH was significantly higher in the ENL group with MN pretreatment.

### 3.4. Pharmacodynamic Evaluation of LidH-Loaded ENLs In Vivo

To test the local anesthetic effect in vivo, we determined the TWL of rats against photothermic stimulation (Figure 5a). Thermal radiation was applied to the central part of the tail of each rat, then the TWL of each rat was measured after different LidH formulation treatments, with or without MN pretreatment (Figure 5b). The TWL of the rats with no treatment on exposure to thermal radiation was used as the blank. The MN/LidH group sustained a better anesthetic effect than the LidH group in the first 25 min. Additionally, TWLs of the MN/ENL groups were both significantly longer than those of MN/LidH group (*p* < 0.001). In general, either MN/Span 80 ENL or MN/Tween 80 ENL application induced a significantly longer and stronger anesthetic effect than that in the LidH group (*p* < 0.001). After approximately 45 min, the TWL measured in the LidH group was similar to that in the blank group, and no anesthetic effect was observed after that, while the TWLs of the MN/ENL groups were extended to more than 80 min.

## 4. Discussion

As a local analgesic, LidH is effective both systemically and topically and several prescriptions and over-the-counter formulations are available, such as creams, gels, and patches [25]. Topical application of LidH has a lower risk of toxicity than systemic administration; thus, it is generally considered safe. The delivery route, thickness of the *stratum corneum*, and the duration of application are three key factors in LidH absorption. LidH needs to penetrate the outer layer of the skin, which consists of keratinized epithelium as a natural barrier, to act when applied topically. Encapsulation of water-soluble LidH in liposomes is a relatively simple but advantageous delivery method, which improves skin penetration ability and maintaining its effect consistently. ENL undergoes much higher vesicle deformability than SLN [26]. Besides vesicular system application, microneedles can compromise the barrier of the *stratum corneum* through a convenient and minimally invasive way and shorten the onset time.

We optimized the ENL preparation method (reverse-phase evaporation method) by adjusting the ratio of SPC to cholesterol, pH of aqueous solution, and sonication process to increase the EE level and cumulative penetration amount of LidH (Appendix A). The organic solvent dissolving lipids was added to the aqueous phase and removed by evaporation. The surfactant is absent in the organic phase of SLN, while Span 80 or Tween 80 helps liposome formation in ENLs. Both particle sizes and PDIs of ENLs are superior to those of the SLN group (Figure 3). The results suggest that ENLs have a more suitable size distribution. The significant difference between the PDI of SLN and that of Tween 80 ENL (*p* < 0.01) could be explained by the lower particle size of Tween 80 ENLs [27]. Span 80 and Tween 80 may form presented with low phase transition temperatures, and thus membrane of ENLs is less rigid with slightly decreased EE; however, the ENLs prevented significant drug waste. [28] Additionally, Tween 80 has a higher hydrophile-lipophile balance (HLB) value than Span 80, causing a smaller EE, as HLB and EE are negatively related [28].

To study cell survival rates, HaCaT cells were exposed to ENL formulations, considering different concentrations, for 6 h (Figure 3d). The results show no cytotoxicity in HaCaT cells with the increase in ENL concentration. Even cultured in medium with an ENL concentration up to 100 μg/mL, the cell viabilities still remained above 99%. The ENL formulations are eligible for in vitro tests of transdermal diffusion and in vivo pharmacodynamics tests.

Regarding the in vitro experiments with both SLN and ENLs, the results of most of them indicate that intact liposomes have quite limited ability to reach the viable epidermis. However, MN is a physical method to promote skin permeation, as it can cross over the *stratum corneum* and even reach the viable epidermis [26]. The ability of MN to penetrate mouse skin as well as the microconduits caused by MN to permit transdermal drug delivery was demonstrated (Figure 4). The number of microconduits displayed on trypan blue-stained skin is 36 after 6 × 6 MN array application, suggesting complete MN penetration. Fluorescence microscopy images clearly show that the *stratum corneum* was pierced and that rhodamine B was delivered through the conduits into the skin. To further prove the function of MN, we conducted transdermal diffusion tests with LidH-loaded SLN, Span 80 ENL, and Tween 80 ENL in vitro, with and without MN pretreatment. All cumulative amounts of LidH that permeated through MN-pretreated skin were higher than those permeating through untreated skin. The permeation profile demonstrated a good capability of MN to improve cutaneous delivery of LidH liposomes.

To enhance skin delivery of LidH, we compared its occlusive and non-occlusive application to the skin (Figure 2b). The absolute delivery amount of LidH was significantly improved under occlusive conditions. Similar improvements have been reported using estradiol- and docetaxel-loaded ENL formulations [26,29]. The *stratum corneum* hyper-hydration caused by occlusion weakens the protective barrier abilities of the skin, which could explain these results to a great extent. Therefore, permeation studies were subsequently performed with occlusive application. Recent data indicate that surfactants enhance the transdermal delivery of LidH-loaded vesicles [12,30,31]. The results of this study showed that ENL formulations deliver significantly more LidH in 6 h than SLN formulations, probably because the surfactants effectively decrease the surface tension at the interface of oil and water, and ENLs can readily deform during diffusion [31]. This characteristic facilitates a deep diffusion of ENL in the skin and LidH release.

The transdermal effect of Tween 80 ENLs was higher than that of Span 80 ENLs after 1 h. Moreover, the particle size of Tween 80 ENLs was lower than that of Span 80 ENLs, whereas the EE was slightly higher (Figure 3). Both of these factors may account for the higher LidH skin penetration during 6 h. However, for local anesthesia, the first hour of LidH transdermal delivery is of high significance, and there was no evident difference in cutaneous drug penetration between the ENL groups in this time range. To shorten the lag time and improve drug permeability, both ENLs were more effective than the LidH solution and SNLs.

The in vivo anesthetic assay was performed to prove the efficacy of MN pretreatment and ENLs as LidH carriers. The results indicate the positive effect of MN array pretreatment (*p* < 0.05). Besides, in accordance with the findings above, the anesthetic effect was prolonged effectively in the groups with the ENLs, suggesting that ENL formulations, especially with Tween 80, facilitate the transdermal permeation of LidH and prolong the anesthetic duration. The properties of biocompatibility and colloidal stability of ENLs may characterize its longer-term anesthesia effect. ENLs may also provide protection against LidH metabolization [32]. 

Through the optimization of the LidH ENL preparations, in vitro drug diffusion tests, and in vivo pharmacodynamic evaluation, the results indicate that the combination of ENLs and MN array pretreatment improves the transdermal delivery of LidH safely and shortens its onset time. Additionally, occlusive application is more suitable for the ENL formulations. It has been reported that 100% of the public and 74% of healthcare professionals positively accept the MN technique as a drug delivery approach [33]. Furthermore, MNs are considered advantageous for drug delivery to children [34].

The preparation procedure can be further optimized to improve the EEs of LidH in ENLs, while improving the delivery efficiency of LidH and its anesthetic effect. In clinical practice, ENL formulations can be sprayed onto the skin upon MN pretreatment. Another comparable approach is the development of LidH-containing dissolvable MNs, which combine the steps of skin treatment and LidH delivery into one step to achieve the rapid onset of local anesthesia, as reported by Yang et al. [35]. Both the methods using the MN pretreatment plus spray and dissolvable MNs are competitive and worthy of further investigation and optimization regarding the safety, efficacy, patient compliance, and manufacture costs.

## 5. Conclusions

The combination of MN pretreatment and ENL formulation exhibits the potential to conduits through the *stratum corneum* with low pain, to improve transdermal lidocaine delivery, shorten the lag time, and to prolong the duration of local anesthetization with negligible cytotoxicity. This strategy has considerable potential for the improvement of patient compliance and for optimization of the efficacy of superficial surgical operations in clinical practice, and can also be utilized for other transdermal formulations in future studies. 

## Figures and Tables

**Figure 1 biomedicines-09-00592-f001:**
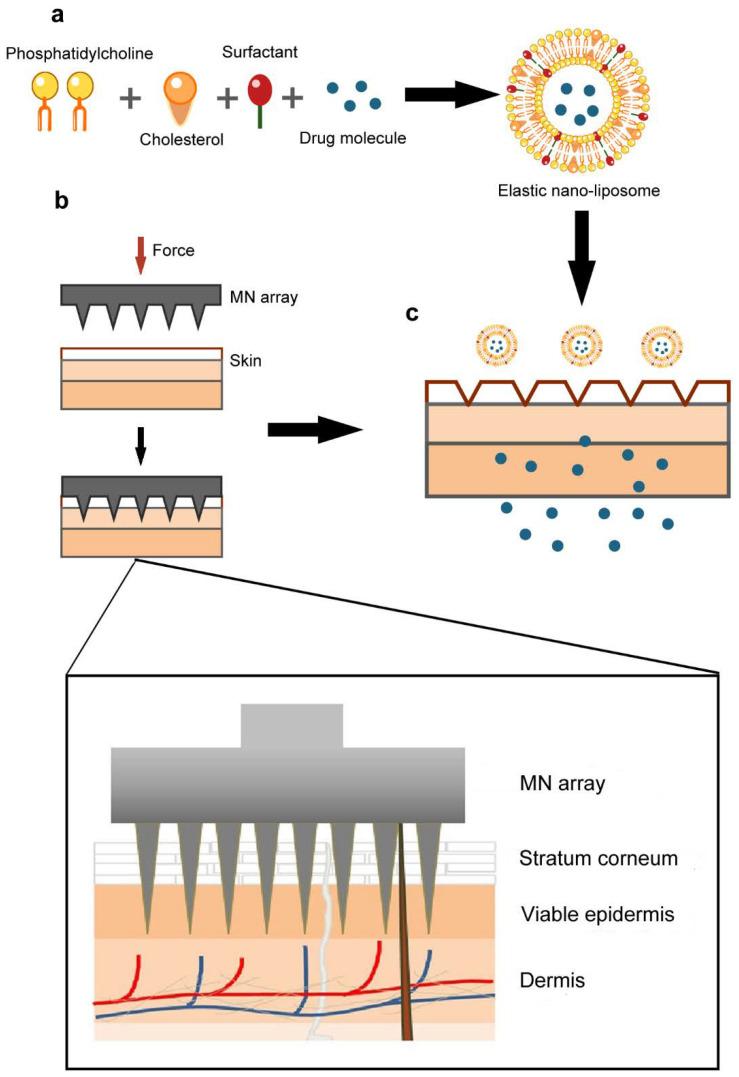
Schematic diagram of research methods employed in the study. (**a**) The ENL consisted of soybean phosphatidylcholine, cholesterol, drug molecule, and surfactant. (**b**,**c**) After treatment, the ENLs containing LidH permeated into skin through the conduits formed. Abbreviations: ENL, elastic nano-liposome; LidH, lidocaine hydrochloride; MN, microneedle.

**Figure 2 biomedicines-09-00592-f002:**
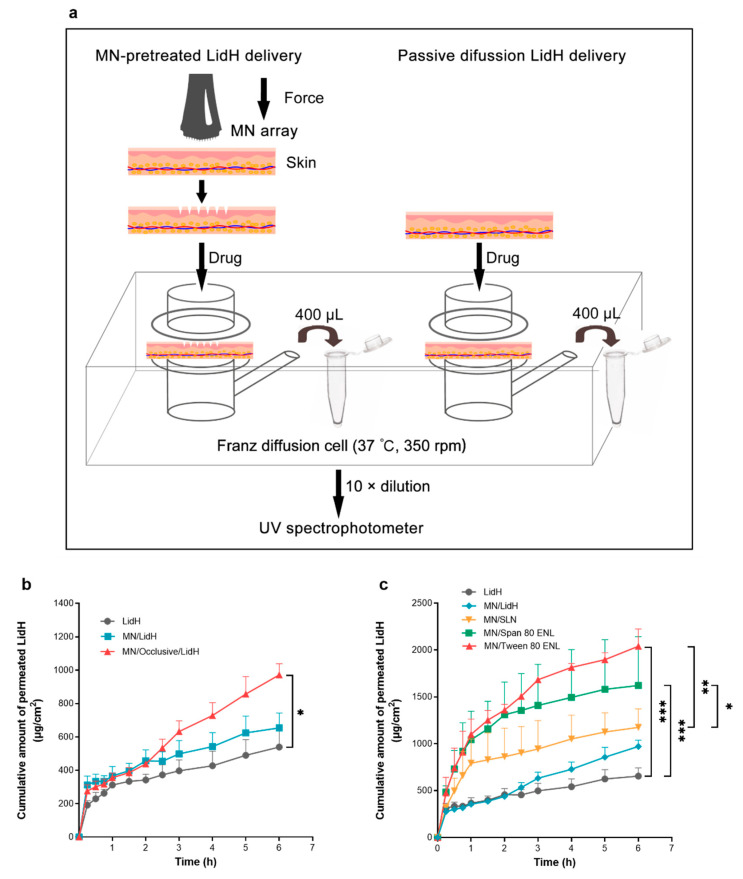
In vitro transdermal delivery of LidH ENL formulations. (**a**) In vitro diffusion studies of LidH ENLs on mouse skin using Franz diffusion cells. (**b**) Cumulative permeation of LidH in MN-pretreated skin after occlusive application. (**c**) Cumulative permeation of LidH in two types of ENL formulation was compared with that in SLN after occlusive application on MN-pretreated skin. Data are shown as mean ± SD, *n* = 3. *: *p* < 0.05, **: *p* < 0.01, ***: *p* < 0.001; Student’s *t*-test. Abbreviations: ENL, elastic nano-liposome; LidH, lidocaine hydrochloride; MN, microneedle; SLN, solid lipid nanoparticle.

**Figure 3 biomedicines-09-00592-f003:**
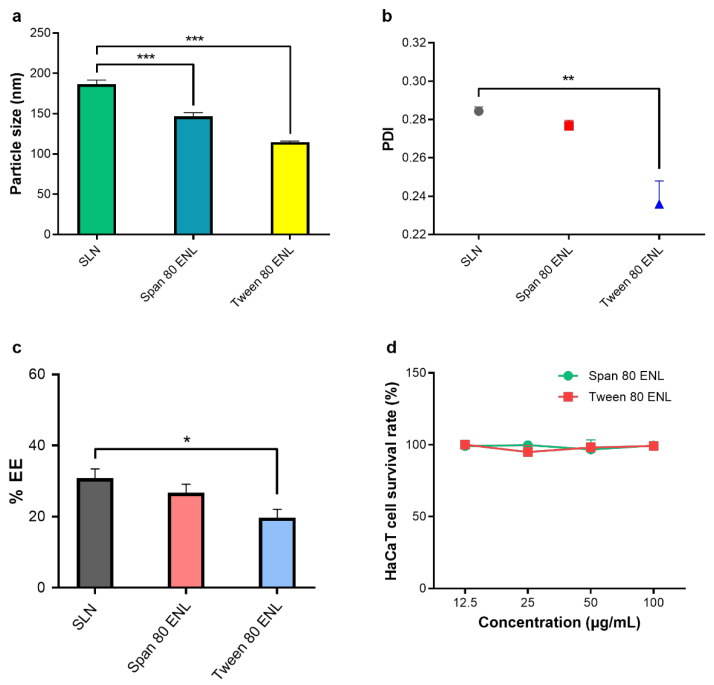
Formulation characteristics. (**a**) Sizes of LidH-loaded liposomes. (**b**) PDIs of LidH-loaded liposomes. (**c**) EEs of LidH-loaded liposomes. Data are shown as mean ± SD, *n* = 3. *, **, and *** indicate statistical significance at *p* < 0.05, *p* < 0.01, and *p* < 0.001 using one-way ANOVA with Bonferroni’s post-hoc test. (**d**) Cell survival rates of HaCaT cells cultured with 12.5, 25, 50, and 100 μg/mL ENL. Cell survival rates were measured as an index of cytotoxicity. Data are shown as mean ± SD, *n* = 3. Abbreviations: SLN, solid lipid nanoparticle; EE, encapsulation efficiency; ENL, elastic nano-liposome; LidH, lidocaine hydrochloride; PDI, polydispersity index.

**Figure 4 biomedicines-09-00592-f004:**
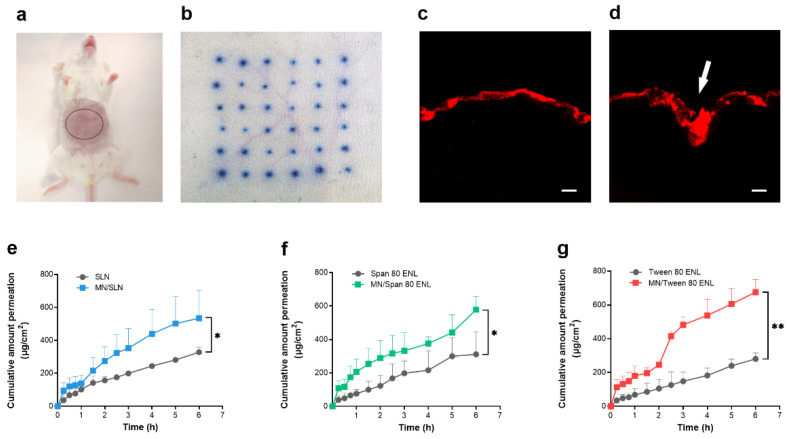
Effective skin penetration of MN and transdermal flux of LidH. Penetration of MN on mice abdominal skin (**a**) with 1% trypan blue staining (**b**). Confocal fluorescence microscopy image showing rhodamine B-stained skin samples with and without MN pretreatment, the white arrow indicates the conduit stained with rhodamine B (**c**,**d**). Scale bar = 100 µm. In vitro transdermal diffusion tests using: (**e**) SLN with and without MN pretreatment, (**f**) Span 80 ENL with and without MN pretreatment, and (**g**) Tween 80 ENL with and without MN pretreatment. Points represent mean ± SD, *n* = 3. *: *p* < 0.05, **: *p* < 0.01; Student’s *t*-test. Abbreviations: ENL, elastic nano-liposome; LidH, lidocaine hydrochloride; MN, microneedle; SLN, solid lipid nanoparticle.

**Figure 5 biomedicines-09-00592-f005:**
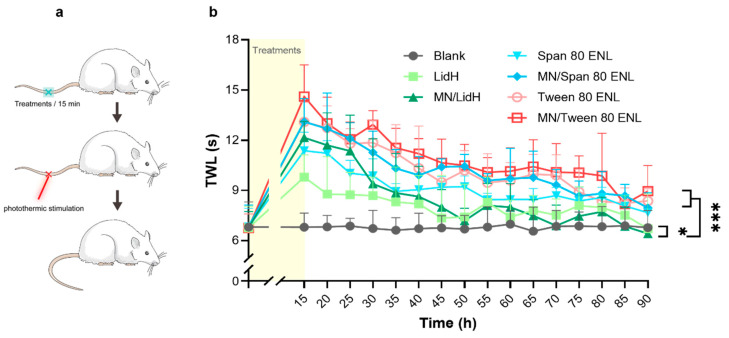
In vivo local anesthetic effect after different treatments with occlusive dressings. (**a**) Schematic diagram of thermal hyperalgesia study on rats. (**b**) Sensitivity of rat tails on stimulation with thermal radiation was tested from 15 min to 1.5 h after LidH or two kinds of ENL treatments with or without MN pretreatment. Data are shown as mean ± SD, *n* = 5. *: *p* < 0.05, ***: *p* < 0.001; two-way ANOVA with Bonferroni’s post-hoc test. Abbreviations: ENL, elastic nano-liposome; LidH, lidocaine hydrochloride; MN, microneedle; TWL, tail withdrawal latency.

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
