# Peer review of "Transdermal Delivery of Lidocaine-Loaded Elastic Nano-Liposomes with Microneedle Array Pretreatment"

_biomedicines, 2021, doi:10.3390/biomedicines9060592_

Round 1
Reviewer 1 Report
The article Liu et al. describes Transdermal Delivery of Lidocaine-Loaded Elastic Nano-lipo-somes with Microneedle Array Pretreatment. The manuscript presented by the authors is interesting and introduces many new elements. Lidocaine hydrochloride (LidH) is a drug that is often transdermally administered, however penetration may be difficult. Authors described study aimed to improve the transdermal delivery of lidocaine hydrochloride (LidH) using elastic nano-liposomes (ENLs) and microneedle (MN) array pretreatment. Then they assessed the anesthetic effects of LidH in various formulations after dermal application were evaluated in vivo in rats.
1) I propose not to repeat the keywords from the title of the manuscript.
General, the manuscript well written. The experiments were well performed and thoroughly analyzed.
I recommend publishing in Biomedicines MDPI.
Author Response
Dear Reviewer:
Re: Manuscript ID: biomedicines-1216815
We are grateful for your comments and suggestion concerning our manuscript “Transdermal Delivery of Lidocaine-Loaded Elastic Nano-liposomes with Microneedle Array Pretreatment” The suggestion is valuable and we have made revision of the keywords in the manuscript.
Thank your for your suggestion again.
Reviewer 2 Report
This article is about a study aimed to improve the transdermal delivery of the anesthetic drug Lidocaine hydrochloride using both nanoparticles and microneedle pretreatment. The nanoparticles used were elastic nano-liposomes (ENLs) and solid lipid nanoparticles (SLNs). This research claims that using this strategy to deliver Lidocaine through the stratum corneum is improving its penetration with low pain and negligible cytotoxicity while shortening the lag time and prolonging the duration of local anesthetization.
The article is well written and easy to understand. An interesting concept is presented and accompanied with convincing results.
Comments:
General comment – The innovation in this article is the use of ENLs and SLN as the carriers of Lidocaine. Thus, there should be a comparison in the introduction to the existed carriers and their performance, in order to understand how ENLs and SLN improve skin penetration, not only comparing to Lidocaine solution.
Lines 128-129 – The measurement of LidH by UV spectrophotometer – in which wavelength does LidH absorb?
Figure 2 b-c – It is not completely clear if the blue curve in b is in non-occlusive conditions? Also, comparing MN\LidH in c to MN\Occlusive\LidH in b they should be the same, but visually doesn't seem the same.
Line 196 – What does it mean "non-sterile bandages"?
Lines 230-232 – The phrase is not well written, they want to describe the effect of the MN on the skin.
Figure 4 – Line 241 – Here is an error in (e) – SLN with and without MN pretreatment.
Lines 249-251 – It is not clear where the numbers are taken from. In figure 2b we can see the opposite trend – in the first hour there is more LidH in the non-occlusive application.
Figure 5b – The pale pink is too bright and can't see well. Enlarge the graph.
Figure 5c – The graphs are not clear, not in good quality. There is no need to show them because these results are clearer in b. It is better to show them in SI, if at all.
Line 284-285 – The sentence is not very clear.
Lines 303-305 – The decrease in EE using ENLs is not very well explained.
Line 327 - There is a mistake in the Figure number, the occlusive\non-occlusive comparison is shown in Figure 2b.
